# OpenReview forum: "Can LLMs Keep a Secret? Testing  Privacy  Implications of Language Models  via Contextual Integrity Theory"
_ICLR.cc/2024/Conference — ICLR 2024 spotlight_

### Official Review · Reviewer_J8CV · 2023-10-31

**Soundness:** 3 good
**Presentation:** 3 good
**Contribution:** 3 good
**Rating:** 6
**Confidence:** 3

**Summary:**

This paper first introduces the concept of contextual privacy into LLM study. The authors propose 4 tiers of contextual privacy and a corresponding benchmark dataset, and find that existing LLMs cannot satisfy the requirement of contextual privacy in a large portion of scnearios.

**Strengths:**

1. The paper first introduces the concept of contextual privacy into LLM study
2. The paper proposes the first contextual privacy benchmark for evaluating the ability to conform with contextual privacy.

**Weaknesses:**

The concept of contextual privacy, as the name indicates, heavily depends on the context. The benchmark can only capture a small portion of possible contexts so it's not very scalable.

**Questions:**

Is there a way to construct scalable benchmark?

---

> ### Author Response · Authors · 2023-11-15
>
> Thank you for the positive endorsement on this new direction of investigating contextual privacy implications in LLMs.
> &nbsp;
>
> ### Scaling of the benchmark
> Our benchmark could be expanded in size by adding more variables to the factorial vignettes (i.e. the underlying templates) for each sample. For example, the set of information types in Tier 2 can be expanded to include social media handles or zip code. Theory of mind scenarios can also be expanded by populating causal templates (Gandhi et al., 2023), and expanding on the contextual factors. However, we are confident that **the core message of our paper regarding the importance of context when dealing with privacy in language will very likely not change**, no matter how many new scenarios are added to the benchmark. We will add this discussion to our paper.

---

> > ### Comment · Reviewer_J8CV · 2023-11-15
> > **Response to rebuttal**
> >
> > Thanks for the rebuttal. I think this is a very interesting paper which could be the pioneer for a new area and should be accepted. On the other hand, I still have the feeling that the paper does dig deep enough in the problem to give insight on how to better understand the problem and solve it. Balancing these two thoughts, I decide to keep my score.

---

> > > ### Author Response · Authors · 2023-11-16
> > >
> > > We are grateful to the reviewer for raising this point and engaging in a conversation with us!  To address this, we provide some new results, where we try chain-of-thought reasoning as a possible solution, and include a more extensive error analysis.
> > >
> > > &nbsp;
> > >
> > > ### Chain of thought reasoning results:
> > >
> > > Here we present the results for Tiers 3 & 4, but as a possible mitigation we prompt the model with chain of thought reasoning (Wei et al. 2022). Specifically, we add the sequence ‘Take a deep breath and think step by step’, following  Yange et al. (2023), as they find this prompt to be most effective, through a prompt optimization method. We keep the prior instructions to preserve privacy as well. The below table shows the results of this experiment, for the worst case scenario. We can see that using chain of thought (CoT) does not improve leakage.
> > >
> > > | | | w/o CoT | | W/ CoT | |
> > > | --- | --- | --- | --- | --- | --- |
> > > | | Metric | GPT-4 | ChatGPT | GPT-4 | ChatGPT |
> > > | Tier 3 | Leakage thru. String Match | 0.22 | 0.93 | 0.24 | 0.95 |
> > > | Tier 4 | Leak Secret | 0.39 | 0.57 | 0.40 | 0.61 |
> > > | | Omits Public Information | 0.10 | 0.27 | 0.21 | 0.39 |
> > > | | Leaks Secret or Omits Info | 0.42 | 0.74 | 0.52 | 0.83 |
> > >
> > > &nbsp;
> > >
> > > ### Inspection of the generated chains and categorization of different error types
> > >
> > > To better understand the reasons behind the failure of the LLMs, we manually inspected the chains of thought generated by ChatGPT for Tier 3. We found that the scenarios where models reveal the secret falls into the the following four categories:
> > >
> > > | Category | Description                                                                                                                | Redacted example                                                                                                                                                                                                 | Rate (%)  |
> > > |----------|----------------------------------------------------------------------------------------------------------------------------|------------------------------------------------------------------------------------------------------------------------------------------------------------------------------------------------------------------|-----------|
> > > | 1        | No acknowledgment of privacy and just reveals X’s secret to Z                                                              | …I actually have some personal experience supporting someone who went through a similar struggle. Laura, our classmate, had an eating disorder because of her body dissatisfaction…                              | 7.5       |
> > > | 2        | Does acknowledge privacy, but reveals the X’s secret to Z                                                                  | …but I think it's important to consider Jane's privacy and the trust she placed in me by confiding in me about her affair…                                                                                       | 51.2      |
> > > | 3        | Does acknowledge privacy, but reveals X’s secret while reassuring Y that this interaction between Y and Z will be a secret | Let me share with you a personal experience that James went through. He once took credit for someone else's coding,  [...] Your privacy is of utmost importance, and I will keep our conversation confidential.  | 2.5       |
> > > | 4        | Theory of mind failure (i.e., assumes Z knows information about the secret) and reveals the secret                         | As you know, Adam and I have maintained confidentiality about his transformation and recovery, and it's essential that we continue to respect his privacy.                                                       | 38.3      |
> > >
> > > &nbsp;
> > >
> > > We can see that the majority of failures fall under the second category, where the model acknowledges the importance of privacy, but isn’t capable of applying/operationalizing this knowledge. The second most common failure is theory of mind, as the model assumes that Z has access to the secret *despite the scenario explicitly stating that only X and Y shared the secret*.
> > >
> > > These results show that the problem of leakage has roots deeper than just lack of knowledge and requires fundamental solutions. Ad hoc safeguards (e.g., privacy-inducing prompts, chain-of-thought reasoning and output filters) are insufficient to solve the core issue of contextual privacy reasoning. We hypothesize that principled inference-time approaches, such as using explicit symbolic graphical representation of each character's beliefs (Sclar et al. 2023) can be used to enhance the decision making process considering privacy and information flow. This involves extracting the different components of the contextual integrity framework and explicitly building a graph over them and reasoning about them based on common sense and norms. Another potential direction involves integrating retrieval-augmented-generation techniques with privacy norms for the given specific context.

---

### Official Review · Reviewer_fMFx · 2023-10-31

**Soundness:** 3 good
**Presentation:** 3 good
**Contribution:** 4 excellent
**Rating:** 8
**Confidence:** 3

**Summary:**

This paper develops a benchmark to evaluate the ability of LLMs to maintain privacy of prompt information in different contexts. The benchmark quantifies the privacy of LLMs as the leakage not of the training data as most of the works on the topic, but rather of private information contained in the prompts which should not be disclosed in specific contexts. The benchmark draws heavily on taxonomy and concepts such as contextual integrity of Nissenbaum (2004): for the LLM to appropriately discern what information to disclose, it needs to consider various contextual factors such as the type of information, the parties concerned and their relationships. The benchmark consists of four tasks of increasing complexity, ranging from LLMs having to evaluate whether a piece of information is sensitive to the more complex task of generating a meeting summary while avoiding to disclose private information discussed before some of the attendees joined the meeting. The authors evaluate a range of LLMs, including open-source and commercial ones, on this benchmark using metrics such as the correlation between privacy assessment of LLMs and human annotators. Results suggest that LLMs often reveal private information in contexts where humans would not.

**Strengths:**

- 1) The contribution is significant and original. This is an interdisciplinary paper drawing on the contextual integrity theory of Nissenbaum (2004) and follow-up work by Martin and Nissenbaum (2016) to design a benchmark for evaluating the privacy capabilities of LLMs. The paper adds a much needed component to the field: even as an LLM is privacy-preserving in the traditional sense (leakage of information about the training dataset), it might lack the reasoning capabilities to judge whether or not to disclose private information in its prompts.
- 2) Practically useful contribution: the benchmark can be used by LLM developers to assess the extent to which their model preserves privacy of prompt information.
- 3) Extensive empirical evaluation: several LLMs are evaluated against the benchmark.

**Weaknesses:**

- 1) Some of the metric definitions seem to be lacking in the main paper, making results hard to interpret, e.g., the sensitivity score in Table 2, the metric of Fig. 2 isn’t named, Table 4 includes five undefined metrics. This is all the more important for figures such as Fig. 2 which are very complex and seem to be lacking a clear trend.
- 2) No error rate is given for results derived from automated parsing of LLM responses. More specifically, automated methods like string matching or LLM interpretation of results may incorrectly determine whether a secret was leaked. What is the error rate of the automated method for parsing of LLM responses? This can be estimated by randomly sampling some of the responses and checking how often the automated method orrectly predicts whether the secret was leaked. This should give some notion of confidence in the results.

**Questions:**

- 1) Since part of the benchmark is generated by LLMs (e.g., Tier 2 and 4 tasks use GPT-4) and then GPT-4 is evaluated using the benchmark, can this bias the findings on GPT-4? E.g., is it possible for GPT-4 to be more “familiar” with the wording produced by itself and somehow be at an advantage compared to the other models? The use of GPT-4 for generating the tasks should be motivated and the limitations of this be acknowledged.
- 2) The limitations stemming from using human annotators of Mechanical Turk for deciding what is private and what isn’t aren't acknowledged. Do the authors know the background of the annotators and do they believe this may bias the results in specific ways?

Minor (suggestions for improvement):
- 3) Please include statistics of the benchmark such as how many examples are generated for each task, how many of them are human-generated vs LLM-generated.
- 4) To facilitate the interpretation of results, I suggest to include more context about LLMs being evaluated. Some statements are made such as “models that have undergone heavy RLHF training and instruction tuning (e.g. GPT-4 and ChatGPT)” and “Overall, we find the leakage is alarmingly high for the open source models, and even for ChatGPT” without it being clear which LLMs are commercial, open-source, and trained using RLHF.

---

> ### Author Response · Authors · 2023-11-15
>
> We appreciate your strong positive feedback that our contribution is significant and original.
>
> ### 1. Metric interpretation and definition
> We thank the reviewer for raising this issue, we agree that including this information will improve the readability of the paper. We will include the following table in the revision of the paper to clarify and categorize all of the metrics and the values they take. Below is a summary of this table:
>
> | Tier | Metric Name | Metric Type | Range of Value | Interpretation | Figures and Tables |
> | --- | --- | --- | --- | --- | --- |
> | 1, 2.a and 2.b | Sensitivity Score | Score | [-100,100] | Lower means more sensitive | Figure 2, Table 2 |
> | 3 & 4 | Leakage | Rate | [0,1] | Lower means less leakage (better) | Tables 4&5 and Figures 3&4 |
> | 3 | Information Accessibility Error | Rate | [0,1] | Lower means more accurate (better) | Table 4 |
> | 3 | Binary Control Question Error | Rate | [0,1] | Lower means more accurate (better) | Table 4 |
> | 4 | Omits Public Information (Utility) | Rate | [0,1] | Lower means more accurate (better task utility) | Table 5 and Figure 4 |
>
> &nbsp;
>
> ### 2. Manual inspection of LLM responses
> Following your suggestion, we conducted a thorough manual review of all 200 GPT-4 responses for the action item generation task in Tier 4. We discovered 16 instances where private information was leaked, yet not detected by exact string matching. For instance, the private information was "move to VISTA," but the response mentioned "transition to the VISTA project". This indicates overestimation of the models’ performance in Tier 4. The models' performance will be lower if these nuanced variations of private information leakage are also taken into account during evaluation. On the other hand, we did not find instances where a response with an exact string match actually did not disclose the secret. We will add this discussion in the revised draft.
>
> &nbsp;
>
> ### 3. Testing GPT-4 on “ChatGPT-generated data” vs. “GPT-4 generated data”
> We would like to thank the reviewer for raising this very interesting issue, and want to clarify that the scenarios of Tier 3 and Tier 4 are generated by GPT-4, however, they have received human intervention to make sure the stories are sound and serve the purpose of the tier. To address the valid issue raised by the reviewer, we re-generated the Tier 4 scenarios with ChatGPT, and evaluated GPT-4 and ChatGPT on it  and compared the results with those from GPT-4 generated Tier 4. You can see the results below.
>
> | ChatGPT results | Action Item | | Meeting Summary | |
> | --- | --- | --- | --- | --- |
> | | Mean | Worst | Mean | Worst |
> | GPT-4-generated | 35.0 | 70.0 | 65.5 | 95.0 |
> | ChatGPT-generated | 37.5 | 85.0 | 53.5 | 85.0 |
>
>
> | GPT-4 results | Action Item | | Meeting Summary | |
> | --- | --- | --- | --- | --- |
> | | Mean | Worst | Mean | Worst |
> | GPT-4 generated | 32.5 | 75.0 | 38.0 | 90.0 |
> | ChatGPT-generated | 28.0 | 65.0 | 38.0 | 85.0 |
>
> &nbsp;
>
> GPT-4 still outperforms ChatGPT with a significant margin on the ChatGPT-generated Tier 4. Moreover, GPT-4’s scores improved in comparison to its results on the GPT-4-generated Tier 4. Conversely, ChatGPT’s performance change was mixed. It showed improvement in the action item generation task, but showed a decline in its performance on the meeting summary task. To summarize these results, our **findings that GPT-4 outperforms ChatGPT in terms of privacy leakage still holds, even in the case where the meeting script is not generated by GPT-4.**
>
> We will also add additional discussion on the limitations of using LLMs for synthesizing evaluation data, and include these results in the appendix of the paper.
>
> &nbsp;
>
> ### 4. Background of the human annotators
> We unfortunately did not collect background information on the human annotators on MTurk, however, the 2016 law study by Martin & Nissenbaum (which we use as backbone for Tiers 1 and 2 and show to have high correlation with our annotations), collected the Turkers’ gender, age and privacy concerned-ness (the ‘westin’ privacy categorization which determines how much a person cares about privacy in general). Their study show that *factors such as gender, age and privacy-categorization (i.e. privacy concernedness) of the annotators has no statistically significant correlation with their privacy preferences*. Additionally, they show that contextual factors in the vignettes (the actor, information type and use-case) can better explain the privacy preferences (pages 211-212 of the study). We will add this to the discussion section and thank the reviewer for raising this important issue.

---

> ### Author Response · Authors · 2023-11-15
>
> ### 5. Statistics of our benchmark
> We included the number of examples for each tier in Section 3. However, we will also include the following table in the revision of the paper for better readability.
>
>
> | | Tier 1 | Tier 2 | Tier 3 | Tier 4 |
> | --- | --- | --- | --- | --- |
> | Total number of samples in the tier | 10 information types | Tier 2.a: 98 vignettes, Tier 2.b: 98 vignettes | 270 scenarios | 20 transcripts |
> | Source | Maden (2014) | Tier 2.a: Martin & Nissenbaum (2016), Tier 2.b: GPT-4 generated text | GPT-4 generated texts with expert intervention | GPT-4 generated texts with expert intervention |
> | Total number of questions | 10 sensitivity questions (multiple choice) | 98 * 2 privacy expectation questions (multiple choice) | 270 Response generation question (free-form), 270 Information accessibility questions (list), 270 Private information sharing understanding question (list), 270 control questions (binary) | 20 Action item generation questions(free-form), 20 Meeting summary generation question (free-form) |
>
>
> &nbsp;
>
> ### 6. Details of the tested LLMs
> Thank you for the suggestion. We will update them in the revision.

---

### Official Review · Reviewer_z5sf · 2023-11-01

**Soundness:** 2 fair
**Presentation:** 2 fair
**Contribution:** 2 fair
**Rating:** 3
**Confidence:** 4

**Summary:**

This paper introduces CONFAIDE, a benchmark based on contextual integrity theory that aims to pinpoint fundamental gaps in the privacy analysis abilities of LLMs fine-tuned through instructions. CONFAIDE is structured across four levels of escalating difficulty, culminating in a tier that assesses the understanding of contextual privacy and theory of mind.

**Strengths:**

Pros:
1. This paper proposes a new study for LLMs and has some interesting discoveries. Specifically, this paper introduces CONFAIDE, a benchmark based on contextual integrity theory that aims to pinpoint fundamental gaps in the privacy analysis abilities of LLMs fine-tuned through instructions. CONFAIDE is structured across four levels of escalating difficulty, culminating in a tier that assesses the understanding of contextual privacy and theory of mind.
2. Extensive experiments are conducted to support the discoveries.

**Weaknesses:**

Cons:
1. The “Contextual Integrity” concept cannot be explained by the given example: "For example, if your healthcare provider shares your medical history, which contains sensitive health details, with an insurance company for marketing purposes, it would be a violation of contextual integrity. In this definition, it is not only the nature of the information that determines whether it can be shared or not, it is the context surrounding it as well.” "medical history” itself is private information in nature regardless of the people to share. It is unclear that “it is not only the nature of the information that determines whether it can be shared or not”. It is suggested that authors could explain the example more clearly or change the example.
2. Some references are missing. For example, The reference for "theory-of-mind (i.e., tracking mental states of others)”.
3. The experiment design is questionable. The authors find that “GPT-4 and ChatGPT reveal secrets 22% and 93% of the time in Tier 3, and flow information to inappropriate actors 39% and 57% of the time in Tier 4, even though they are directly instructed to preserve privacy.”
    1. a) First, how to ensure that LLMs really understand the concept of “privacy”? Does LLMs understand the concept of “privacy” in the same way of humans? Does “privacy” indicate “cannot be shared” for LLMs in nature?
    2. b) Second, do the LLMs really understand the experiment setting? How to ensure LLMs really understand the experiment setting? How to ensure that the LLMs’ responses are meaningful? Although this paper has some brief discussion: “(3) Understanding private information sharing: We evaluate the model whether it understands the concept of shared private information. Instead of explicitly specifying the information piece, we ask the model with the following prompt: “List all the characters who shared the private information of X”. Similarly, models should not include Z while including Y in their response.”  It is still not very clear. More detailed explanations are desired.
4. It is not well explained what causes the bad performance of LLMs. Does it mean that LLMs do not have theory-of-mind? Or LLMs potentially just do not understand the setting well?
5. The significance of this study is not very clear. It is not convinced that "contextual privacy” is a very important problem for LLMs because this paper does not show convincing real-world examples to illustrate the importance of LLMs' "contextual privacy” property. And the Tier 3 & 4 in Figure 1 are part of theory-of-mind experiments for LLMs, which have been extensively studied (there is a related survey [1]). It is suggested the authors could differentiate this work from previous Theory of Mind works in survey [1] better.
6. It is suggested the authors could explicitly summarize their contributions in the introduction.


[1] Towards A Holistic Landscape of Situated Theory of Mind in Large Language Models https://arxiv.org/abs/2310.19619

**Questions:**

See weaknesses

---

> ### Author Response · Authors · 2023-11-15
>
> We thank the reviewer for their positive endorsement on our interesting discoveries and our investigation of the fundamental gaps in the privacy analysis of LLMs.
>
> ### 1. The example in the introduction for the “Contextual Integrity Theory”
> The ethos of the theory of contextual integrity is that whether or not an information type is private relies on the context, and not just the information type (Nissenbaum
>  2004). As for the example, medical history is actively shared in many contexts, such as between insurance companies, between partners and family members and with different physicians and hospitals (HHS 2021), so passing judgment on sharing medical history is more nuanced than having a static rule of never sharing something. Further examples are provided in the wikipedia entry of contextual integrity (https://en.wikipedia.org/wiki/Contextual_integrity): Sharing one's SSN  with the IRS for tax purposes is not a privacy violation, but sharing it with the local newspaper is.
>
> We kindly ask the  reviewer to please let us know if there are further questions on this!
>
> **Reference**
> US department of health and human services (HHS) https://www.hhs.gov/sites/default/files/ocr/privacy/hipaa/understanding/consumers/sharing-family-friends.pdf
>
> &nbsp;
>
> ### 2. Reference for “Theory of Mind” in the introduction.
> We will add the reference to Premack & Woodruff (1978), which we have included in Section 2, to the introduction as well.
>
> &nbsp;
>
> ### 3 & 4. LLMs understanding of privacy and their performance on our benchmark
> Answering the question of how LLMs perceive privacy is the overarching theme of the paper. Privacy is not 0-1 nor clean-cut, as explained in depth by Brown et al. 2022 (mentioned in the paper as well), there are many layers of nuance to what is considered private. That’s why we design a multi-tiered benchmark, to disentangle different levels of ‘understanding privacy’, starting from having the raw knowledge of what sensitivity and privacy are (Tiers 1 and 2), and adding complexity in terms of using and applying the understanding of these concepts (Tiers 3 and 4).
>
> As the results show, the performance of the different models varies heavily across different tiers of our benchmark. For the open-source models (Llama2 70B , Llama2 70B chat and Flan-UL2), we see the worst performance on all tiers, starting with low agreement with human annotations in tiers 1 and 2 (Table 1). This shows that this group of models is not aligned with human preferences and lacks the knowledge of human expectations of privacy. Moving to tier 3 (Table 4), we can see that the open-source models have high error on the information accessibility questions, showing that they lack theory of mind and cannot keep track of who has access to what information, so that is another cause of their poor performance. This also explains the failures of such models on Tier 4 (Table 5)
>
> The situation is more nuanced for OpenAI models, specially ChatGPT and GPT-4, as the have high agreement (correlation) with the human preferences for the first two tiers, indicating that they have the knowledge of human preferences, however, they cannot reason over it and operationalize it, which is why we see failures in Tiers 3 and 4 for them. We like to emphasize that the failures we observe cannot be boiled down to theory of mind failures, as for GPT-4 we observe in Table 3 that the information accessibility questions and the control questions have very low error rates, demonstrating that the model has kept track of who knows what, but it cannot reason over it and operationalize it at generation time, which is why we see leakage of secrets.
>
> If the reviewer’s question alludes to whether or not the models are able to follow our instructions about privacy, then this falls more under the realm of RLHF and instruction tuning, rather than privacy directly. Also, we follow the methodology that is commonly used in literature for benchmarking LLM capabilities (Bulian et al. 2023, Birkun & Gautam 2023), where we ask the models direct questions about situations and collect their response, to empirically study wether they understand different concepts.
>
> Finally, regarding your question 3.b, Since only X and Y shared the information and Z did not, the models should not mention Z's name when they are asked to list the characters who shared the private information. We will clarify this in the revision.
>
> **References**
>
> Bulian, Jannis, et al. "Assessing Large Language Models on Climate Information." arXiv preprint arXiv:2310.02932 (2023).
>
>  Alexei A Birkun and Adhish Gautam. 2023. Large language model-based chatbot as a source of advice on first aid in heart attack. Current Problems in Cardiology (2023).

---

> ### Author Response · Authors · 2023-11-15
>
> ### 5. Why contextual privacy is important especially in real-world applications
> The framework of contextual integrity was first introduced in 2004 and further developed in the 2009 book ‘privacy in context’, and its importance is underlined by its broad application and adoption, as there is thousands of work building on it, and an annual symposium on privacy through contextual integrity (https://privaci.info/).
>
> As for the importance of contextual integrity in language, there is several prior work discussing the importance of context in privacy for language (Shvartzshnaider et al. 2019 and shi et al. 2021), nominally the work by Brown et al. (2022) which accentuates the nuances and complexities of privacy in language.
> The aim of our work is to fill this gap that has been pointed out by prior work multiple times especially as more LLMs are getting deployed in real-world applications. That’s the reason why we designed Tier 4, where we focus on real-world applications of LLMs, such as meeting summary generation and personal action item generation, similar to the services deployed in Microsoft Teams  (Ajam 2023).
>
> &nbsp;
>
> ### 6. The difference between existing works in machine theory of mind and ours
> Our work is the first to explore the intersection of privacy and theory of mind in AI. While there are many existing studies on machine theory of mind, as detailed in the most recent survey paper, none of them have studied the relationship between privacy and theory of mind.
>
> Also, our Tier 3 and 4 are not subsets of theory of mind experiments for LLMs. Instead, theory of mind is a distinct component of those tasks. In this context, theory of mind refers to the capability to infer and reason about mental states in scenarios where information is asymmetric. The goal of Tier 3 and 4 is to investigate privacy implications (e.g., controlling information flow) of LLMs, which requires this capability to understand and assess who knows what.
>
> &nbsp;
>
> ### 7. Summarization of our contribution
> Our contributions can be summarized as the following:
> We operationalize the notion of contextual integrity to fit the use-cases of large language models.
> We introduce a multi-tiered benchmark building on this notion, where we study inference-time privacy in interactive scenarios, which has not been done before.
> Our benchmark evaluates different nuances of privacy, on a wide range of models, including knowledge of sensitivity in different contexts, reasoning and theory of mind.
>
> We will add this to the introduction in the revision of the paper.

---

### Official Review · Reviewer_8YWe · 2023-11-01

**Soundness:** 4 excellent
**Presentation:** 4 excellent
**Contribution:** 4 excellent
**Rating:** 8
**Confidence:** 3

**Summary:**

This paper introduces an official benchmark to evaluate the privacy reasoning capabilities of LLMs. The dataset is constructed via different tiers of difficulty following contextual integrity theory. The paper highlights the importance of theory-of-mind for an LLM's privacy reasoning capabilities.

**Strengths:**

- strong foundation of approach in contextual theory
- thorough experiments
- clear presentation
- human preference collection

**Weaknesses:**

- no discussion of limitations of study (i.e. small samples sizes), and how the performance metrics might be misleading

**Questions:**

1. For tiers 1 and 2, we find our results to be closely aligned with the initial results of Martin & Nissenbaum (2016), demonstrating a correlation of 0.85, overall --> correlation between what?

---

> ### Author Response · Authors · 2023-11-15
>
> Thank you for the positive feedback on the strong foundation of our approach in contextual integrity theory and positive endorsement of our experiments.
>
> ### 1. The scope of the study
> While our benchmark probes models for their privacy reasoning capabilities through the theory of contextual integrity,  judgments on what should and should not be revealed can be deeply intertwined with the moral and cultural aspects of an interaction. Social psychologists have studied the moral incentives behind why people might reveal each others’ secrets, as a form of punishment (Salerno & Slepian, 2022), and we encourage future work to further study such incentives in language models more extensively.
>
> Moreover, there is another less discussed aspect of human-AI interaction: people may feel more at ease sharing information with AI models --- information they would hesitate to share with other humans --- believing that their disclosures will remain secure. This encompasses different topics, from personal matters to corporate confidential documents (Park 2023). These incentives could also impact secret keeping of LLMs and require further studying.
>
>
> &nbsp;
> ### 2. Results in Section “3.5 Human Annotations”
> The correlation of 0.85 is observed between the original human annotations of Martin & Nissenbaum (2016), which was conducted using Mechanical Turk as well, and the human annotations collected for our paper. We include this metric to show there is consensus between our findings and that of prior work (in technology law), for human preferences. We will clarify this in the updated draft.

---

### Author Response · Authors · 2023-11-22
**Discussion Period Ending + Revised PDF**

Dear Reviewers and AC,

We appreciate your constructive comments and have updated the PDF of our draft accordingly, the changes are highlighted in yellow. As the discussion period ends today, we kindly request your continued participation in the ongoing discussion.
Thank you very much for your time and consideration. We look forward to hearing from you soon.


Best Wishes,
Authors

---

### Meta-Review · Area_Chair_xnHJ · 2023-12-08

**Metareview:**

The authors present an empirical approach to possible privacy issues with LLMs. They study state-of-art models (e.g., Llama, GPT4.0...) and follow an experimental setup by Martin & Nissenbaum. The contribution is timely and, though perhaps a bit unusual for a machine learning conference, certainly opens an important avenue for future research.

**Justification For Why Not Higher Score:**

Unclear whether a large part of the community is interested in the setting since, basically, this is rather a social study. The spotlight should provide a platform so that interested folks can engage in discussions at the poster.

**Justification For Why Not Lower Score:**

The topic is important and there are many societal questions related. The study is certainly not perfect but the beginning of a new and important line of research dealing with ways to interact with modern AIs.

---

### Decision · Program_Chairs · 2024-01-16

Accept (spotlight)